# The Impact of Cyberbullying Victimization on Academic Satisfaction among Sexual Minority College Students: The Indirect Effect of Flourishing

**DOI:** 10.3390/ijerph20136248

**Published:** 2023-06-29

**Authors:** Jeoung Min Lee, Jinhee Park, Heekyung Lee, Jaegoo Lee, Jason Mallonee

**Affiliations:** 1School of Social Work, Wichita State University, Wichita, KS 67260, USA; 2College of Education, Auburn University, Auburn, AL 36849, USA; 3College of Education, California State University, Sacramento, CA 95819, USA; 4School of Social Work, Jackson State University, Jackson, MS 39217, USA; 5College of Health Sciences, The University of Texas at El Paso, El Paso, TX 79968, USA

**Keywords:** cyberbullying victimization, flourishing, academic satisfaction, LGBTQ college students

## Abstract

This study examines the association between cyberbullying victimization and academic satisfaction through flourishing (psychological well-being) among 188 LGBTQ college students utilizing the lens of general strain theory and positive psychology. Results indicate that flourishing as a mediator explains the association between cyberbullying victimization and academic satisfaction among LGBTQ college students. For these students, flourishing can serve as a protective factor for their academic satisfaction. This finding highlights the need for college counselors, faculty, and administrators to foster psychological well-being among cyberbullied LGBTQ college students. Practice implications will guide the development of a campus-wide cyberbullying intervention for these students.

## 1. Introduction

Social networking sites (SNS) have become an increasingly common forum for youth and college students to maintain their social relationships or for leisure and entertainment. They use email, text messages, Zoom, Google Hangouts, or Microsoft Teams more frequently than during any other time in their lives. Increasing time spent in cyberspace can lead to potential risks of involvement in cyberbullying [1]. While a universal definition of cyberbullying has yet to be agreed upon, existing studies conceptualize that cyberbullying occurs when people repeatedly harm others in cyberspace, which is a subtype of aggressive behavior using email, SNS, and communication apps through various electronic devices [2].

According to the National Center for Educational Statistics (NCES), cyberbullying now occurs more often than traditional bullying in public schools in the United States [3]. Sexual-minority (i.e., LGBTQ) students are vulnerable, and they may easily become targets for victimization in cyberspace. Social identities (e.g., cultural affiliation, sexual orientation, race/ethnicity, gender, or disability) are socially constructed identities often associated with social bias, stereotyping, and discrimination in our society, which may produce bias-based bullying, the extension of which in digital platforms is bias-based cyberbullying [4,5]. Studies have found that LGBTQ individuals are more likely to be bullied and harassed in cyberspace, which is closely linked with a higher risk of suicidality, poorer mental health, and negative academic outcomes when compared to heterosexual students [6,7]. A systematic literature review across 27 empirical studies revealed that the percentage of cyberbullying among LGBTQ youth is between 10.5% and 71.3%; moreover, LGBTQ students reported higher levels of cyberbullying experiences than their heterosexual counterparts [7].

In positive psychology, *flourishing* is considered an essential element to increasing individuals’ well-being. In other words, the amount of flourishing in one’s life connects with an individual’s experience of a “good life” [8]. Although the nature of flourishing consists of multidimensional aspects, most researchers agree that flourishing includes but is not limited to components of well-being—happiness and life satisfaction [9]. The current study conceptualizes psychological well-being as the outcome of flourishing [10]. The existing literature has established that a college degree benefits individuals both economically and socially, including attaining better jobs, lower unemployment, higher salaries and work benefits, less crime, more advanced knowledge, and better health and life expectancy [11,12]. Although the achievement of flourishing and academic satisfaction are among the most critical tasks for college students, few studies have examined the impact of cyberbullying on psychological and academic satisfaction for students who identify as sexual minorities. In order to add to the literature base, this study employs general strain theory and positive psychology to explore these relationships.

Applying general strain theory [13], this study focuses on the negative effect of LGBTQ college students’ cyberbullying victimization experiences on academic satisfaction, and the potential role of flourishing as a mediator of this impact. Cyberbullying victimization experiences are harmful and stressful events during college life, and LGBTQ college students experience cyberbullying victimization as a strain that stirs negative emotions. These emotions might negatively impact their academic life, resulting in dissatisfaction with academic achievement, their major, or their school. Although prior studies have shown that the strain that bullied adolescents feel leads to poor academic performance [14,15], cyberbullied LGBTQ college students’ academic satisfaction has not been examined.

According to positive psychology, when individuals establish a positive meaning of their life and purpose, they can flourish, which improves their quality of life [16]. A recent study found that cyberbullied college students with higher levels of a sense of purpose in life have lower levels of depressive symptoms [17]. If LGBTQ college students with cyberbullying victimization experiences exhibit higher levels of flourishing, they may experience less severe problems with academic satisfaction.

### 1.1. Cyberbullying and LGBTQ College Students’ Academic Satisfaction

Academic satisfaction is a core positive mental health indicator and a critical domain of subjective well-being among college students [18]. However, the academic satisfaction of LGBTQ college students has not been considered frequently in cyberbullying research. Some studies have examined whether bullying experiences impact academic achievement for adolescents [19,20,21], but chiefly by looking at grades [20,22]. For example, students with higher levels of bullying performed poorly from an academic perspective, resulting in lower grades [22]. In cyberbullying research specifically, poor academic adaptation and achievement by college students has also been reported [22,23,24]. Cyberbullying negatively impacts academic achievement through decreased concentration in studying and lower academic motivation, interest, and commitment to school, resulting in underachievement or failure [25]. For example, a study in Israel found that undergraduate students who were cyberbullied via email experienced poor academic adaptation, such as decreased academic ability, lower motivation to learn, and loss of general satisfaction with the academic environment [23]. Compared to academic performance by cyberbullied heterosexual high school students, cyberbullied non-heterosexual counterparts reported higher victimization and were more likely to receive failing academic grades. Given the dearth of research on the impact of cyberbullying victimization on LGBTQ students’ academic satisfaction, this study uniquely contributes to the cyberbullying literature. 

### 1.2. Effect of Flourishing on Cyberbullying and Academic Satisfaction in LGBTQ College Students

While extant research has demonstrated a relationship between cyberbullying and psychological well-being in youth and adolescents [26,27,28,29], only a handful of studies convey how cyberbullying impacts LGBTQ college students specifically. Studies have found cyberbullying victimization to be negatively associated with the psychological well-being and academic performance of LGBTQ college students. A systematic review indicated that LGBTQ students who experienced cyberbullying commonly presented with psychological and emotional problems (e.g., depression, lower self-esteem, suicidal ideation and attempts), behavioral problems, and poor academic achievement (e.g., lower GPAs) [7]. Additional research found that LGBTQ college students who experienced cyberbullying reported greater depressive symptoms when compared to heterosexual students who similarly experienced cyberbullying [30]. Furthermore, a study conducted in China revealed that cyberbullied LGBTQ college students can develop social anxiety [31]. A sample of 5730 LGBTQ secondary school students in the United States who experienced bullying victimization in a school setting also experienced lower academic achievement and lower self-esteem [32]. This research suggests that bullying experienced by LGBTQ college students influences both their psychological well-being and academic satisfaction. Although research establishing a relationship between psychological well-being and academic success in LGBTQ college students is scarce, one study discovered that LGBTQ college students consistently reported severe mental health issues and frequent academic challenges [33]. Other researchers found that LGBTQ students who experienced bullying victimization showed decreased well-being, such as depression and low self-esteem, and that their poor psychological well-being also negatively impacted their academic achievement [34]. 

### 1.3. The Current Study

The current study examines the potential associations between cyberbullying and academic satisfaction for LGBTQ college students, and how this association may be explained by levels of flourishing. Specific hypotheses include: 

**Hypothesis 1.** LGBTQ college students who have experienced cyberbullying victimization will report low levels of academic satisfaction.

**Hypothesis 2.** LGBTQ college students who have experienced cyberbullying victimization will report low levels of flourishing, and those students with low levels of flourishing will report low levels of academic satisfaction. 

## 2. Materials and Methods

### 2.1. Participants and Data Collection Procedures

This is a cross-sectional study with a convenience sample of college students from four universities in midwestern, south central, and southern U.S. regions. After gaining an approval letter from the Institutional Review Board of the author’s affiliated institution, the investigators recruited college students from the universities through the dissemination of a flyer via school email that included study information, a confidentiality explanation, and a consent statement. Data were collected with a self-administered online survey using Qualtrics and QuestionPro from November 2022 to January 2023. To be eligible to participate in this study, participants were required to be at least 18 years old. After the students’ agreement, participants could begin the survey voluntarily. When they completed the survey, they were able to enter raffles for compensation. A total of 878 students completed the survey, and a subsample of 188 LGBTQ-identified college students was selected for this study. Participant ages ranged from 18 to 48 years old (*M* = 22.5). The composition of the selected students included 146 (77.6%) females, 33 (17.6%) males, and 9 (4.8%) nonbinary/third gender/prefer not to say. The students classified themselves as 121 (64.4%) White, 19 (10%) African American, 21 (11.2%) Hispanic, 15 (8%) Asian, and 12 (6.4%) others.

### 2.2. Measures

The dependent variable, *college students’ academic satisfaction*, was measured using the Satisfaction with Academics Scale from the College Student Subjective Well-being Questionnaire [18]. The Satisfaction with Academics Scale included six items, such as “I really enjoy my classes”, “I am happy with my academic major”, “Overall, my experiences in my classes have been excellent”, “I have had a great academic experience at my university”, “I am happy with how I’ve done in my classes”, and “I am satisfied with my academic achievement since coming to my university”. Response options were on a Likert-type scale, including 1 = Strongly disagree, 2 = Disagree, 3 = Slightly disagree, 4 = Mixed or neither agree nor disagree, 5 = Slightly agree, 6 = Agree, and 7 = Strongly agree. The Cronbach’s alpha for this measure was 0.90.

The independent variable, *cyberbullying victimization*, included nine items measured from the Cyberbullying and Online Aggression Survey Instrument [2]. Example items are “I have been cyberbullied in the last 30 days”, “Someone posted mean or hurtful comments about me online in the last 30 days”, “Someone posted a mean or hurtful picture online of me in the last 30 days”, and “Someone pretended to be me online and acted in a way that was mean or hurtful in the last 30 days”. Responses were measured on a 4-point Likert-type scale, including 1 = Never, 2 = Once, 3 = A few times, and 4 = Many times. The Cronbach’s alpha for this measure was 0.92.

The mediator for the study was *flourishing*, including eight items, which were assessed using the Flourishing Scale [35]. Example items are “I lead a purposeful and meaningful life”, “My social relationships are supportive and rewarding”, “I am engaged and interested in my daily activities”, and “I actively contribute to the happiness and well-being of others”. Responses were measured on a 7-point Likert-type scale, including 1 = Strongly disagree, 2 = Disagree, 3 = Slightly disagree, 4 = Mixed or neither agree nor disagree, 5 = Slightly agree, 6 = Agree, and 7 = Strongly agree. The Cronbach’s alpha for this measure was 0.91. 

The covariate for this study included age, which was a continuous variable. Sociodemographic variables were utilized, including biological sex, race/ethnicity, and employment status, but these were not found to be significantly correlated with the primary study variables and were excluded for the purpose of analysis. 

### 2.3. Statistical Analyses

Descriptive statistics and correlation analyses were conducted to describe the variables’ characteristics and correlations. Also, to examine the direct and indirect relationships (Hypotheses 1 and 2), the PROCESS macro Version 4.1 [36] in SPSS 27 (IBM Corp, New York, NY, USA) was utilized. The bias-corrected and accelerated (BCa) bootstrap interval was used to examine the indirect association, which corrects bias and skewness in the distribution of the bootstrap estimates [37].

## 3. Results

Table 1 displays bivariate correlations among the study variables. Academic well-being was positively correlated with flourishing (*r* = 0.46, *p* < 0.01) and age (*r* = 0.16, *p* < 0.05), while cyberbullying victimization was negatively correlated with flourishing (*r* = −0.23, *p* < 0.01).

Table 2 and Figure 1 show the direct effects. Cyberbullying victimization was not significantly associated with academic satisfaction after controlling for age covariates (*C′*: *β* = −0.14, *ns*), while cyberbullying victimization was significantly associated with flourishing (a: *β* = −1.60, *p* < 0.01). Flourishing was significantly associated with academic satisfaction (b: *β* = 0.38, *p* < 0.001). Table 2 demonstrates a significant effect of cyberbullying victimization on academic satisfaction through flourishing; indirect effect (ab) = −0.61, Bca: CI = [−1.20, −0.06]. The total effect (*C* = *C′* + ab; *β* = −0.75, *ns*) includes the direct effect (*C′*: *β* = −0.14, *ns*) and the indirect effect (ab): *β* = −0.61, Bca: CI = [−1.20, −0.06] after controlling for all covariates. Results show that flourishing as a mediator fully explained the association between cyberbullying victimization and academic satisfaction among LGBTQ college students, which reflects a full mediation model.

## 4. Discussion

Cyberbullying victimization experiences can have significant adverse effects on physical and psychological well-being among college students, especially those from marginalized populations, including sexual and gender minorities. The current study examined the mediating role of flourishing (well-being) in the relationship between cyberbullying victimization experiences and perceived academic satisfaction among LGBTQ college students. Findings from this study contribute to the existing literature by examining how variables emphasized in positive psychology, such as students’ well-being, can function as a protective factor in LGBTQ college students’ experiences of cyberbullying victimization. 

Flourishing, which encompasses the concept of subjective and psychological well-being [38], is related to one’s self-perception as thriving, doing well, and perceiving life as good [39]. It is not surprising to find the detrimental effect of cyberbullying victimization experiences on students’ psychological and subjective well-being, which is also confirmed by previous studies. For example, in a study conducted with gay and bisexual men during emerging adulthood in Taiwan, the authors found that cyber harassment victimization was significantly associated with lower quality of life [40]. Additionally, a study in Turkey showed that Turkish university students reported significantly lower levels of psychological well-being when they experienced cyberbullying victimization [41]. A study conducted with adolescents in the United Kingdom found that cyberbullying victimization was significantly associated with poorer mental well-being [42].

This study’s findings reveal that cyberbullying victimization experiences were not significantly associated with academic satisfaction. This result was inconsistent with previous studies showing that cyberbullied students reported significantly lower academic performance, such as lower academic grades or school attachment [7,15]. Although a previous study indicated that the level of academic performance is directly related to students’ perceived academic satisfaction [43], our finding did not confirm this link. Future research may be needed to clarify such associations between cyberbullying victimization, academic satisfaction, and actual academic performance. Another possible explanation could be related to intersectionality [4,44,45]. For example, Angoff and Barnhart [4] posited that individuals with multiple intersecting minority identities might be at a greater risk of bullying victimization. Their study findings also supported the intersectionality framework demonstrating significant within-group variations (i.e., gender, race, and sexual orientation) in the study participants’ bullying and cyberbullying experiences and their academic performance (grades).

Our findings show that flourishing fully mediates the association between cyberbullying and academic satisfaction. A greater number of victimization experiences had an indirect negative impact on students’ perceived academic success via diminished well-being. Such results are consistent with previous studies confirming the role of well-being in the relationship between cyberbullying victimization and academic and mental health outcomes [34,43]. Members of the LGBTQ community tend to be more frequently prejudiced, stereotyped, and discriminated against in our heterosexist society [46], and they more often become the target of various types of hate crimes on- and offline [47,48]. Such ongoing exposure to victimization experiences through various forms of bullying might lead to their feeling a lack of satisfaction or fulfillment in their lives, which in turn has a negative impact on adaptive functioning in their academic environment. On the other hand, students possessing a higher level of flourishing, that is, demonstrating good mental and psychological well-being by showing their ability to cope with life stressors, may be able to better overcome negative consequences of life difficulties such as cyberbullying victimization, which might reduce its impact on their mental health and academic outcomes [49]. Such an indirect path informs service providers about the significant need to intervene to enhance the well-being of LGBTQ college students so that they can become more resilient in the face of adversity and challenges in life [38].

## 5. Conclusions

### 5.1. Limitations and Implications for Research

The findings from this study should be understood considering the following limitations. As a result of the cross-sectional research design through an online self-report survey, results from the current study cannot investigate causal inferences between cyberbullying victimization, flourishing, and academic satisfaction. Future research should consider a longitudinal study design to further examine the effects of cyberbullying victimization on LGBTQ college students’ flourishing and academic satisfaction over time. Also, participants were recruited from just a few institutions in three regions of the United States; therefore, sample characteristics may not be reflective of the diverse range of students throughout the entire U.S. college student population. Generalizability is further limited due to convenience sampling. Furthermore, the students’ sexual minority status was categorized into one cultural group within the current study, which might not have captured the diverse and unique experiences among students within the LGBTQ community. Their other identities, such as gender, race/ethnicity, and socioeconomic status, might also have influenced their experiences of bullying victimization and well-being different ways. Therefore, future research should include a larger and more diverse sample of LGBTQ college students and examine variations among different intersecting identities. Although the previous literature indicated that students’ perceived academic success was significantly correlated with actual academic performance [50], utilizing a self-report measure to examine students’ perceived level of academic performance and success may not accurately capture their actual academic achievement. Future research should not only utilize instrumentation examining college students’ academic satisfaction, but also include an objective measure of gauging students’ actual academic performance, such as their overall GPA. 

Although previous studies have focused on well-being among individuals experiencing cyberbullying victimization [40,41], the concept of flourishing is a relatively recent construct being introduced in the literature [51], which is a higher level of well-being encompassing the components of happiness, life satisfaction, meaning, engagement, purpose in life, and personal growth [38]. Future research efforts, such as examining various factors influencing flourishing for victims of cyberbullying and identifying effective intervention programs that can help increase the level of flourishing, may provide valuable information in supporting LGBTQ students who were cyberbullied in coping with the associated negative consequences. When developing intervention research, students’ intersecting multiple identities should be considered since different identities an individual holds may interact and manifest as privilege and oppression experienced in their daily life, which can become either protective or risk factors [4]. Seeing individuals’ victimization experiences through an intersectional lens will allow researchers to also consider unique cultural factors that can help shape an understanding of self, others, and the world, which will impact psychological and academic well-being. Future studies incorporating individuals’ unique cultural values influenced by their interlocking identities might provide valuable insight into prevention and intervention programs that help cyberbullied individuals identify more effective coping strategies to deal with adversity [52]. 

### 5.2. Practice Implications

It is well established that students who identify as LGBTQ already experience disproportionately lower rates of well-being when compared to their heterosexual peers and those who conform to traditional gender role expectations [53,54]. Students who identify as LGBTQ are more prone to experience cyberbullying during what may be the most important transition period in their lives (e.g., coming out), further supporting the need for cyberbullying intervention efforts that target the unique experiences and challenges faced by LGBTQ students [55]. Within this context, this study’s findings carry significant practice implications for potentially mediating the negative impacts of cyberbullying on LGBTQ college students’ academic satisfaction. Recognizing that the well-being of LGBTQ college students is compounded by the distress associated with experiences of systemic discrimination, oppression, and violence [56], intervention efforts should consist of a combination of individual-level and collaborative campus-wide efforts that promote inclusivity and offer resources specifically for LGBTQ students [7]. 

At the same time, these efforts should recognize that LGBTQ identities exist alongside, and interact with, other identities these students may possess [4,44,45]. Framing both individual and campus inclusivity and anti-bullying efforts through the lens of intersectionality, which recognizes that addressing each point of a person’s identity individually is limited when compared to addressing how multiple points of identity interact [57], colleges and universities will be better positioned to address LGBTQ student needs more holistically and comprehensively. Intervention efforts tailored for LGBTQ students should be sustained over time [56], and by incorporating intersectionality theory, also address other points of identity where they may experience discrimination and marginalization in both in-person and online environments [4,44,45]. 

This study’s findings suggest that a focus on strengthening psychological well-being and fostering a sense of flourishing may mediate the relationship between cyberbullying victimization and academic satisfaction. Interventions aimed at doing so may be most effective if LGBTQ students who are victims of cyberbullying are identified early and provided with immediate support [28]. To support academic success and well-being for these students, the negative impacts of cyberbullying need to be addressed through interventions that promote protective factors, including psychological well-being. This finding is consistent with previously identified protective factors of increasing self-esteem [58]; strengthening resilience [59]; and fostering emotional openness, positive identity, and a sense of hope [60]. This study uniquely adds to the limited literature base on intervention efforts specifically for LGBTQ students experiencing cyberbullying [55] by providing evidence of the potential for promoting psychological well-being to offset some of the negative impacts of cyberbullying and lead to increased academic satisfaction.

## Figures and Tables

**Figure 1 ijerph-20-06248-f001:**
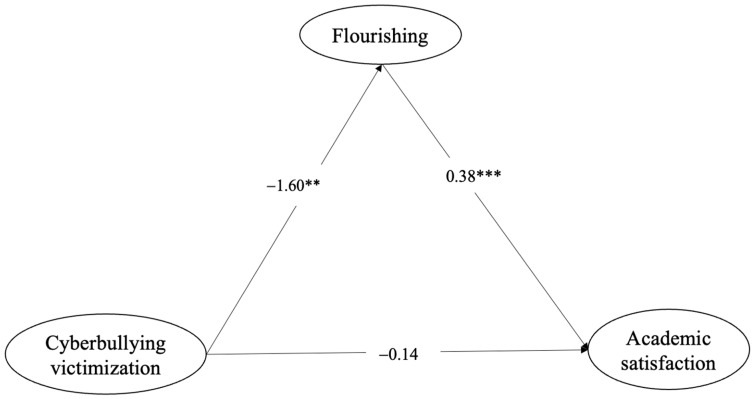
The indirect effect of flourishing on the relationship between cyberbullying victimization and college students’ academic satisfaction. ** *p* < 0.01, *** *p* < 0.001.

**Table 1 ijerph-20-06248-t001:** Descriptive statistics and correlations among the study variables.

Variables	M (SD)	1	2	3	4
1. Academic satisfaction	29.27 (8.13)	-			
2. Cyberbullying victimization	9.31 (1.35)	−0.12	-		
3. Flourishing	40.08 (9.60)	0.46 **	−0.23 **	-	
4. Age	22.52 (4.83)	0.16 *	0.00	0.01	-

* *p* < 0.05. ** *p* < 0.01.

**Table 2 ijerph-20-06248-t002:** The Indirect effect of cyberbullying victimization on academic satisfaction.

Predictor (X)	Mediator (M)	Outcome (Y)	Effect of X on M (a)	Effect of M on Y Controlled by X (b)	Total Effect C	Direct Effect C′	Indirect Effect	
							a × b	95% Bca CI
Cyberbullying victimization	Flourishing	Academic satisfaction	−1.60 **	0.38 ***	−0.75	−0.14	−0.61	−1.20 −0.06
			covariance	Age	0.26 *	0.27 *		

* *p* < 0.05; ** *p* < 0.01; *** *p* < 0.001. Bca (bias-corrected and accelerated) interval addresses bias and skewness in the distribution of the bootstrap estimates.

## Data Availability

Not applicable.

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
