# Peer review of "The Impact of Cyberbullying Victimization on Academic Satisfaction among Sexual Minority College Students: The Indirect Effect of Flourishing"

_ijerph, 2023, doi:10.3390/ijerph20136248_

Round 1
Reviewer 1 Report
This is an interesting paper which makes a useful contribution. In general it is clearly written, with a well structured literature review, and good presentation of results and discussion. I have minor suggestions for revision.
Lines 32-33 make it clear that you are referring to the USA here
Lines 107-111 ‘low’ or ‘lower’ levels of academic well-being is imprecise – what is the reference point? Lower than what?
Line 128 give details of the age distribution of the sample
Lines 168-170 the numbers here simply repeat Table 1! We don’t need both – maybe shorten the text? What struck me about Table 1 was the relatively high correlation (.46**) between academic wellbeing and flourishing – perhaps deserves more comment.
Lines 216-217 possibly the inconsistency is due to the sample being LGBTQ in this study?
Author Response
This is an interesting paper which makes a useful contribution. In general it is clearly written, with a well structured literature review, and good presentation of results and discussion. I have minor suggestions for revision.
Comment 1:
Lines 32-33 make it clear that you are referring to the USA here
We added “in the United States” on line 35
Comment 2:
Lines 107-111 ‘low’ or ‘lower’ levels of academic well-being is imprecise – what is the reference point? Lower than what?
We changed from “lower” to “low” on line 136
Comment 3:
Line 128 give details of the age distribution of the sample
We added participant’s age ranges on lines 151-152
Comment 4:
Lines 168-170 the numbers here simply repeat Table 1! We don’t need both – maybe shorten the text? What struck me about Table 1 was the relatively high correlation (.46**) between academic wellbeing and flourishing – perhaps deserves more comment.
We removed the description of descriptive statistics on lines 195-197.
We made the change from academic well-being to academic satisfaction for clarification. Academic satisfaction was measured using the Satisfaction with Academics Scale from the College Student Subjective Well-being Questionnaire. It is mainly focused on academic satisfaction; also, flourishing and academic satisfaction correlation score was .46, so it was not highly correlated.
Comment 5:
Lines 216-217 possibly the inconsistency is due to the sample being LGBTQ in this study?
Yes, our study’s LGBTQ samples showed a different result, and we also provided another potential factor on lines 254-259.
Reviewer 2 Report
I thank the authors for this submission. The content is well-written and discusses an engaging topic. I appreciate the authors' hard work in attempting to address their research inquiries. Consequently, I offer my suggestions concerning two central areas that require improvement.
(A) Literature review
The literature review of the manuscript needs to be more robust. While the authors made reference to a systematic review [5] in the introduction, they failed to incorporate insights from this work throughout the submission. To enhance the manuscript, I strongly encourage the authors to integrate insights from the following works into their literature review and discussions:
References:
- Angoff, H. D., & Barnhart, W. R. (2021). Bullying and cyberbullying among LGBQ and heterosexual youth from an intersectional perspective: findings from the 2017 National Youth Risk Behavior Survey. Journal of school violence, 20(3), 274-286.
- Gower, A. L., Rider, G. N., McMorris, B. J., & Eisenberg, M. E. (2018). Bullying victimization among LGBTQ youth: Critical issues and future directions. Current sexual health reports, 10, 246-254.
- Doxbeck, C., & Noel, T. K. (2023). I felt truly powerless: Narrative research on cyberbullying victimization and negative outcomes in graduate education. International Journal of Educational Research Open, 4, 100245.
- Hinduja, S., & Patchin, J. W. (2020). Bullying, cyberbullying, and LGBTQ students. Cyberbullying Research Centre, 11990, 2073.
(B) the concept of digital intersectionality
In addition, the authors overlooked the concept of intersectionality in their discussion of victimization and sexuality in digitalized society. I strongly urge the authors to employ an intersectional lens in discussing cyberbullying and victimization.
References:
- Lazarus, S. (2019). Just married: the synergy between feminist criminology and the Tripartite Cybercrime Framework. International Social Science Journal, 69(231), 15-33.
- Tynes, B.M., Schuschke, J. & Noble, S.U., 2016. Digital intersectionality theory and the black matter movement. In: Noble, S.U. and Tynes, B.M., eds. The intersectional internet: race, sex, class, and culture online. Berlin: Peter Lang International Academic Publishers.
By incorporating insights from these works (Lazrus, 2019; Tynes, Schuschke, & Noble, 2016) and adopting a digital intersectional perspective, the manuscript will be strengthened and provide a more comprehensive analysis of the topic.
The digital intersectionality will provide valuable insights into how cyberbullying victimization affects academic well-being by considering individuals' intersecting identities and experiences.
To illustrate my point, here are some potential values of digital intersectionality in understanding the impact of cyberbullying victimization on academic well-being:
- Multidimensional analysis: Intersectionality allows researchers to examine the unique experiences and challenges faced by individuals who may hold multiple marginalized identities. By considering intersecting identities such as race, gender, sexuality, disability, and socioeconomic status, researchers can better understand how these factors intersect with cyberbullying victimization and impact academic well-being. This multidimensional analysis helps capture the complexity of individuals' experiences.
- Uncovering differential effects: Intersectionality enables researchers to uncover differential effects of cyberbullying victimization on academic well-being based on intersecting identities. By recognizing these differential effects, researchers can identify and address disparities and systemic biases that may influence academic outcomes. For example, an intersectional analysis may reveal how cyberbullying affects individuals differently depending on their race or socioeconomic status.
- Identifying unique challenges: Intersectionality helps identify the unique challenges that individuals with intersecting marginalized identities may face in the context of cyberbullying victimization. For instance, cyberbullying may be compounded by discrimination based on not only sexuality (LGBTQ group) but also socioeconomic, class, gender, and race, leading to additional psychological distress and academic obstacles. By understanding these unique challenges, researchers can develop targeted interventions and support systems that address the specific needs of different groups.
- Considering cultural factors: Intersectionality encourages researchers to consider cultural factors that may influence how cyberbullying victimization affects academic well-being. Cultural beliefs, norms, and values can shape individuals' responses to cyberbullying and impact their academic experiences. By recognizing the influence of culture, the authors can develop culturally sensitive strategies to address the academic impact of cyberbullying victimization.
- Contextualizing coping strategies: Intersectionality helps researchers understand how individuals with intersecting marginalized identities may employ coping strategies to navigate the academic impact of cyberbullying victimization. Cultural, social, and identity-related factors can influence coping mechanisms. By considering intersectional experiences, researchers can identify effective coping strategies and develop interventions that promote resilience and academic success.
- Promoting inclusive policies and practices: Intersectionality informs the development of inclusive policies and practices that support individuals affected by cyberbullying victimization in academic settings. By recognizing students' intersecting identities and experiences, researchers can advocate for policies that address different groups' unique needs and challenges. This can include implementing comprehensive anti-bullying measures, providing resources for mental health support, and fostering inclusive and supportive school climates.
In a nutshell, I strongly believe the lens of digital intersectionality will bring significant value to understanding how cyberbullying victimization affects academic well-being by offering a multidimensional analysis, uncovering differential effects, identifying unique challenges, considering cultural factors, contextualizing coping strategies, and promoting inclusive policies and practices. By adopting an intersectional lens, the authors can contribute to a more nuanced understanding of the academic impact of cyberbullying victimization and develop interventions that support the diverse needs of affected individuals.
While the content is well-written and discusses an engaging topic, however, there are still some linguistic errors present. The manuscript will still benefit from thorough proofreading and editing.
Author Response
I thank the authors for this submission. The content is well-written and discusses an engaging topic. I appreciate the authors' hard work in attempting to address their research inquiries. Consequently, I offer my suggestions concerning two central areas that require improvement.
(A) Literature review
The literature review of the manuscript needs to be more robust. While the authors made reference to a systematic review [5] in the introduction, they failed to incorporate insights from this work throughout the submission. To enhance the manuscript, I strongly encourage the authors to integrate insights from the following works into their literature review and discussions:
References:
- Angoff, H. D., & Barnhart, W. R. (2021). Bullying and cyberbullying among LGBQ and heterosexual youth from an intersectional perspective: findings from the 2017 National Youth Risk Behavior Survey. Journal of school violence, 20(3), 274-286.
- Gower, A. L., Rider, G. N., McMorris, B. J., & Eisenberg, M. E. (2018). Bullying victimization among LGBTQ youth: Critical issues and future directions. Current sexual health reports, 10, 246-254.
- Doxbeck, C., & Noel, T. K. (2023). I felt truly powerless: Narrative research on cyberbullying victimization and negative outcomes in graduate education. International Journal of Educational Research Open, 4, 100245.
- Hinduja, S., & Patchin, J. W. (2020). Bullying, cyberbullying, and LGBTQ students. Cyberbullying Research Centre, 11990, 2073.
Thank you so much for providing resources to make our paper robust.
To address a review’s suggestion, we added some sentences in the introduction and discussion by using of some of the suggested articles on lines 39-43, 56-59, and 304-321.
(B) the concept of digital intersectionality
In addition, the authors overlooked the concept of intersectionality in their discussion of victimization and sexuality in digitalized society. I strongly urge the authors to employ an intersectional lens in discussing cyberbullying and victimization.
References:
- Lazarus, S. (2019). Just married: the synergy between feminist criminology and the Tripartite Cybercrime Framework. International Social Science Journal, 69(231), 15-33.
- Tynes, B.M., Schuschke, J. & Noble, S.U., 2016. Digital intersectionality theory and the black matter movement. In: Noble, S.U. and Tynes, B.M., eds. The intersectional internet: race, sex, class, and culture online. Berlin: Peter Lang International Academic Publishers.
By incorporating insights from these works (Lazrus, 2019; Tynes, Schuschke, & Noble, 2016) and adopting a digital intersectional perspective, the manuscript will be strengthened and provide a more comprehensive analysis of the topic.
The digital intersectionality will provide valuable insights into how cyberbullying victimization affects academic well-being by considering individuals' intersecting identities and experiences.
To illustrate my point, here are some potential values of digital intersectionality in understanding the impact of cyberbullying victimization on academic well-being:
- Multidimensional analysis: Intersectionality allows researchers to examine the unique experiences and challenges faced by individuals who may hold multiple marginalized identities. By considering intersecting identities such as race, gender, sexuality, disability, and socioeconomic status, researchers can better understand how these factors intersect with cyberbullying victimization and impact academic well-being. This multidimensional analysis helps capture the complexity of individuals' experiences.
- Uncovering differential effects: Intersectionality enables researchers to uncover differential effects of cyberbullying victimization on academic well-being based on intersecting identities. By recognizing these differential effects, researchers can identify and address disparities and systemic biases that may influence academic outcomes. For example, an intersectional analysis may reveal how cyberbullying affects individuals differently depending on their race or socioeconomic status.
- Identifying unique challenges: Intersectionality helps identify the unique challenges that individuals with intersecting marginalized identities may face in the context of cyberbullying victimization. For instance, cyberbullying may be compounded by discrimination based on not only sexuality (LGBTQ group) but also socioeconomic, class, gender, and race, leading to additional psychological distress and academic obstacles. By understanding these unique challenges, researchers can develop targeted interventions and support systems that address the specific needs of different groups.
- Considering cultural factors: Intersectionality encourages researchers to consider cultural factors that may influence how cyberbullying victimization affects academic well-being. Cultural beliefs, norms, and values can shape individuals' responses to cyberbullying and impact their academic experiences. By recognizing the influence of culture, the authors can develop culturally sensitive strategies to address the academic impact of cyberbullying victimization.
- Contextualizing coping strategies: Intersectionality helps researchers understand how individuals with intersecting marginalized identities may employ coping strategies to navigate the academic impact of cyberbullying victimization. Cultural, social, and identity-related factors can influence coping mechanisms. By considering intersectional experiences, researchers can identify effective coping strategies and develop interventions that promote resilience and academic success.
- Promoting inclusive policies and practices: Intersectionality informs the development of inclusive policies and practices that support individuals affected by cyberbullying victimization in academic settings. By recognizing students' intersecting identities and experiences, researchers can advocate for policies that address different groups' unique needs and challenges. This can include implementing comprehensive anti-bullying measures, providing resources for mental health support, and fostering inclusive and supportive school climates.
In a nutshell, I strongly believe the lens of digital intersectionality will bring significant value to understanding how cyberbullying victimization affects academic well-being by offering a multidimensional analysis, uncovering differential effects, identifying unique challenges, considering cultural factors, contextualizing coping strategies, and promoting inclusive policies and practices. By adopting an intersectional lens, the authors can contribute to a more nuanced understanding of the academic impact of cyberbullying victimization and develop interventions that support the diverse needs of affected individuals.
We appreciate your description of the digital intersectionality theory, which is a critical theory for current and future studies. The digital intersectionality framework is a brand-new framework for us. We learned a lot from you and you provided significant insight for our future study.
We added the explanation of bias-based cyberbullying in the introduction part on lines 39-43, and the intersectionality framework is addressed in the discussion and research/practice implications sections on lines 254-259, 304-321, and 337-347.
Comments on the Quality of English Language
While the content is well-written and discusses an engaging topic, however, there are still some linguistic errors present. The manuscript will still benefit from thorough proofreading and editing.
The editor reviewed this paper.
Round 2
Reviewer 2 Report
I'm pleased that you found my previous comments helpful in enhancing your manuscript. The manuscript has thus been sufficiently enhanced to merit publication in IJERPH, following a minor correction.
I suggest a minor revision due to the following issues. Some portions of the manuscript where you referenced "intersectionality" (lines 240, 323-326, and 329-331) require citations/references, such as the recent papers on intersectionality I suggested when recommending the intersectionality lens to you.
I am aware that you cite [56] at line 326, but this source is more than 30 years old, published in 1991, and does not address digital offenses and victimizations in cyberspace. Therefore, it is advantageous to supplement this 1991 reference with more recent works, such as those previously suggested.
Moderate editing of text required
Author Response
I'm pleased that you found my previous comments helpful in enhancing your manuscript. The manuscript has thus been sufficiently enhanced to merit publication in IJERPH, following a minor correction.
I suggest a minor revision due to the following issues. Some portions of the manuscript where you referenced "intersectionality" (lines 240, 323-326, and 329-331) require citations/references, such as the recent papers on intersectionality I suggested when recommending the intersectionality lens to you.
We included the 4,44,45 citations in the lines the reviewer indicated and added a new reference:
- Tynes, B.; Schuschke, J.; Noble S.U. Digital intersectionality theory and the #BlackLivesMatter movement. In The Intersectional Internet: Race, Sex, Class, and Culture Online; Noble, S.U., Tynes, B.M., Eds.; Peter Lang Publishing, Inc.: New York, NY USA, 2016; pp. 21-40.
Please double check if the format is correct.
Yes, we did.
I am aware that you cite [56] at line 326, but this source is more than 30 years old, published in 1991, and does not address digital offenses and victimizations in cyberspace. Therefore, it is advantageous to supplement this 1991 reference with more recent works, such as those previously suggested.
We replaced a recent version and changed the citation number to 57.
​​57. Crenshaw, K. W. Mapping the Margins: Intersectionality, Identity Politics, and Violence Against Women of Color. In Foundations of Critical Race Theory in Education, Routledge: England UK, 2023, pp.73-307.